# OpenReview forum: "Adversarial Contrastive Decoding: Aligning Large Language Models via Exploiting Their Safety and Harm"
_ICLR.cc/2025/Conference — ICLR 2025 Conference Withdrawn Submission_

### Official Review · Reviewer_oNeV · 2024-10-25

**Soundness:** 3
**Presentation:** 2
**Contribution:** 3
**Rating:** 5
**Confidence:** 4

**Summary:**

The paper Adversarial Contrastive Decoding: Aligning Large Language Models via Exploiting Their Safety and Harm introduces a method called Adversarial Contrastive Decoding (ACD) to improve the safety alignment of large language models (LLMs). ACD works by optimizing two opposing soft system prompts: the Safeguarding Prompt (SP), which promotes safe outputs, and the Adversarial Prompt (AP), which encourages unsafe outputs. These prompts are used during decoding to create a contrast between safe and harmful responses, allowing the model to better align with safety objectives without retraining.

**Strengths:**

+ The paper introduces a creative method, Adversarial Contrastive Decoding (ACD), which optimizes opposing soft prompts (Safeguarding and Adversarial) to align large language models with safety objectives.
+ The authors conduct a wide range of experiments across multiple benchmarks and models, providing comprehensive evidence of ACD's effectiveness in improving safety without sacrificing general performance.

**Weaknesses:**

- The experiment should incorporate evaluations using jailbreak attack methods to demonstrate the performance improvements of the proposed approach.
- The dependence on a manually generated anchor dataset for Opposite Prompt Optimization may introduce biases due to the quality of the generated samples. If the anchor data does not sufficiently cover a broad range of harmful content, the effectiveness of ACD could be constrained.
- Further clarification or discussion on certain questions is needed.

**Questions:**

Evaluating the Harmlessness Rate (HLR) with ChatGPT may lead to an unfair assessment, as it risks a high rate of false positives. A combined approach using rule-based methods alongside LLM judgment might be more effective.

---

### Official Review · Reviewer_WWwc · 2024-11-04

**Soundness:** 2
**Presentation:** 3
**Contribution:** 2
**Rating:** 5
**Confidence:** 4

**Summary:**

This paper presents Adversarial Contrastive Decoding (ACD), a prompt-based framework to enhance the safety of Large Language Models (LLMs) without requiring extensive retraining. The approach involves the generation of two contrasting prompts: a Safeguarding Prompt (SP) designed to promote safe outputs and an Adversarial Prompt (AP) to expose the model’s capacity for harmful responses. By leveraging Opposite Prompt Optimization (OPO), the authors propose to align LLMs more safely at the inference stage using minimal computational overhead. Experimental results on several benchmarks and models suggest that ACD improves model safety and reduces susceptibility to jailbreak attacks.

Pros:
+ ACD achieves safety alignment without extensive retraining, offering a lightweight and resource-friendly alternative
+ ACD reduces the success rate of jailbreak attacks, indicating some robustness in security-sensitive contexts

Cons:
- The reliance on prompt-based contrastive decoding rather than deeper alignment mechanisms limits ACD’s potential depth in safety enhancement, as it may fail to generalize across varied contexts
- The framework introduces an extra layer of complexity with dual prompts and specific parameter settings (e.g., the contrastive coefficient α), which may limit practical scalability, especially for larger models or diverse use cases
- The decoding-time safety methods for LLMs are not new, and many of them can be discussed, such as [1-3]
- The study primarily uses benchmark tests without extensive exploration of diverse, real-world applications, which raises concerns about the method’s practical applicability and robustness

Ref:
[1] Alon G, Kamfonas M. Detecting language model attacks with perplexity[J]. arXiv preprint arXiv:2308.14132, 2023.
[2] Phute M, Helbling A, Hull M D, et al. Llm self defense: By self examination, llms know they are being tricked[C]//The Second Tiny Papers Track at ICLR 2024. 2023.
[3] Zhang Y, Ding L, Zhang L, et al. Intention analysis prompting makes large language models a good jailbreak defender[J]. arXiv preprint arXiv:2401.06561, 2024.

**Strengths:**

+ ACD achieves safety alignment without extensive retraining, offering a lightweight and resource-friendly alternative
+ ACD reduces the success rate of jailbreak attacks, indicating some robustness in security-sensitive contexts

**Weaknesses:**

- The reliance on prompt-based contrastive decoding rather than deeper alignment mechanisms limits ACD’s potential depth in safety enhancement, as it may fail to generalize across varied contexts
- The framework introduces an extra layer of complexity with dual prompts and specific parameter settings (e.g., the contrastive coefficient α), which may limit practical scalability, especially for larger models or diverse use cases
- The decoding-time safety methods for LLMs are not new, and many of them can be discussed, such as [1-3]
- The study primarily uses benchmark tests without extensive exploration of diverse, real-world applications, which raises concerns about the method’s practical applicability and robustness

Ref:
[1] Alon G, Kamfonas M. Detecting language model attacks with perplexity[J]. arXiv preprint arXiv:2308.14132, 2023.
[2] Phute M, Helbling A, Hull M D, et al. Llm self defense: By self examination, llms know they are being tricked[C]//The Second Tiny Papers Track at ICLR 2024. 2023.
[3] Zhang Y, Ding L, Zhang L, et al. Intention analysis prompting makes large language models a good jailbreak defender[J]. arXiv preprint arXiv:2401.06561, 2024.

**Questions:**

See weakness

---

### Official Review · Reviewer_vdEZ · 2024-11-04

**Soundness:** 2
**Presentation:** 2
**Contribution:** 2
**Rating:** 3
**Confidence:** 4

**Summary:**

The paper introduces a contrastive decoding method that directs the model to produce safe outputs by distinguishing between harmful and safe logits. These logits are obtained by adding corresponding soft prompts before the query. Experiments demonstrate that this method can reduce harmfulness across different models.

**Strengths:**

1. The prompt optimization method proposed in the paper is general and novel.

2. Experiment results show that the method performs well across models and datasets.

**Weaknesses:**

1. Experiment results are not comprehensive. (1) The harmfulness of models seems to be evaluated only through direct attacks. However, many jailbreaking methods, including GCG [1], PAIR [2], and AutoDAN [3], should also be assessed for their applicability. (2) There is a lack of analysis regarding the decoding cost of the proposed methods compared to other decoding techniques. While it is expected that the decoding time and resource usage may be higher, the extent of this increase requires further analysis. (3) Generation capability is evaluated using AlpacaEval and TruthfulQA, which lack generality. Important skills such as reasoning and understanding are not evaluated using more specific NLP datasets like GSM8K, SQUAD, and XSum.

2. The analysis of the decoding method lacks solidity. Figure 5 illustrates the impact of $\alpha$ on models' harmfulness, yet its effect on the models' general capabilities is not discussed. This omission leads to a fundamental gap in understanding why the method might work. While reducing unsafe logits is expected to enhance model safety, both safe and unsafe logits can contribute to general capabilities, especially when queries are unrelated to safety. Consequently, reducing unsafe logits might degrade overall performance. I would appreciate further analysis on this aspect to address my concerns.



[1] Universal and Transferable Adversarial Attacks on Aligned Language Models

[2] Jailbreaking Black Box Large Language Models in Twenty Queries

[3] AutoDAN: Generating Stealthy Jailbreak Prompts on Aligned Large Language Models

**Questions:**

Please refer to the weakness section.

---

### Official Review · Reviewer_Hp4r · 2024-11-05

**Soundness:** 2
**Presentation:** 1
**Contribution:** 2
**Rating:** 3
**Confidence:** 2

**Summary:**

This work introduces Adversarial Contrastive Decoding (ACD), which leverages both safeguarding and adversarial soft prompts in contrastive decoding to enhance the safety of responses generated by large language models (LLMs). Unlike other methods, ACD does not require an additional guiding model; instead, it relies on prompt tuning to update the soft prompt used in contrastive decoding. Once trained, these soft prompts can be reused to generate safe responses. Experimental results demonstrate that the proposed method is effective compared to other baselines.

**Strengths:**

- The use of a soft prompt in contrastive decoding is novel.
- The experimental results suggest that the proposed method seems promising.

**Weaknesses:**

- **Clarity Issues**: The paper is difficult to follow due to undefined and unclear terms. In Figure 2, terms like $D_{HR}$, $D_S$, and $D_{HA}$ are unclear, as is the meaning of each loss term, which complicates the understanding of how safeguarding and adversarial soft prompts are trained. Additionally, terms such as $logit_S$ and $logit_A$ in Equation 5 are undefined.
    - In the Baseline section, terms like nID and oID are also ambiguous, and the meaning of “prompt” remains unclear—specifically, whether it refers to a soft or hard prompt in this context.
- **Insufficient Analysis**: Lines 27-29 of the abstract claim that "ACD only needs a lightweight prompt tuning on a relatively small anchor dataset without training the target model." However, the experiment section lacks any discussion on the size of this anchor dataset, which is critical for understanding the method’s practicality.
    - Additionally, fine-tuning LLMs on the anchor dataset could serve as a strong baseline due to its simplicity and feasibility, yet the paper does not include any comparison with this approach. Was there a specific reason for excluding this as a baseline?
    - Unlike other baselines, the proposed method requires two forward passes during inference, which could reduce its efficiency. However, there is no analysis or discussion of this trade-off, even in the Limitations section.
    - Finally, the paper would benefit from including the baselines used in Table 6 within Table 2. Extensive validation of the proposed method across various models and with existing baselines is essential for a comprehensive evaluation.

**Questions:**

- Consider placing the table captions above the tables for consistency.

---

### Note · Authors · 2025-01-20

I have read and agree with the venue's withdrawal policy on behalf of myself and my co-authors.